

# An experimental study of rill sediment delivery in purple soil, using the volume-replacement method

Yuhan Huang[1], Xiaoyan Chen[1], Banglin Luo[1], Linqiao Ding[1] and Chunming Gong[2]

[1] College of Resources and Environment, Southwest University/Key Laboratory of Eco-environments in Three Gorges Region (Ministry of Education), Southwest University, Chongqing, China
[2] Chongqing Surveying and Design Institute of Water Resources Electric Power and Architecture, Chongqing, China

## ABSTRACT

Experimental studies provide a basis for understanding the mechanisms of rill erosion and can provide estimates for parameter values in physical models simulating the erosion process. In this study, we investigated sediment delivery during rill erosion in purple soil. We used the volume-replacement method to measure the volume of eroded soil and hence estimate the mass of eroded soil. A 12 m artificial rill was divided into the following sections: 0–0.5 m, 0.5–1 m, 1–2 m, 2–3 m, 3–4 m, 4–5 m, 5–6 m, 6–7 m, 7–8 m, 8–10 m, and 10–12 m. Erosion trials were conducted with three flow rates (2 L/min, 4 L/min, and 8 L/min) and five slope gradients (5°, 10°, 15°, 20°, and 25°). The eroded rill sections were refilled with water to measure the eroded volume in each section and subsequently calculate the eroded sediment mass. The cumulative sediment mass was used to compute the sediment concentration along the length of the rill. The results show that purple soil sediment concentration increases with rill length before eventually reaching a maximal value; that is, the rate of increase in sediment concentration is greatest at the rill inlet and then gradually slows. Steeper slopes and higher flow rates result in sediment concentration increasing more rapidly along the rill length and the maximum sediment concentration being reached at an earlier location in the rill. Slope gradient and flow rate both result in an increase in maximal sediment concentration and accumulated eroded amount. However, slope gradient has a greater influence on rill erosion than flow rate. The results and experimental method in this study may provide a reference for future rill-erosion experiments.

## INTRODUCTION

Soil erosion is a serious problem in China. Researching the soil erosion process and controlling soil and water loss important to improving human survival in the environment. It is also important for the sustainable development of poor areas in China especially (*Lei,*

Corresponding author
Xiaoyan Chen, c400716@126.com

*Zhang & Yan, 2009*; *Lei et al., 2008*). In soil erosion, rill erosion contributes significantly to water and soil loss on sloping farmland (*Li, Zhu & Li, 2008*; *Bhattarai & Dutta, 2007*; *Miao et al., 2014*; *Kröpfl et al., 2013*). In regions with purple soil, rill erosion is an important mechanism of erosion on hillslopes, and is the main source of sediment particles in the watershed. Rill erosion therefore has a potentially important effect on the development and evolution of drainage-area landforms (*Cai, Zhu & Wang, 2004*; *Nearing, Norton & Bulgakov, 1997*; *Miao, Ni & Borthwick, 2010*; *Miao et al., 2015*). In recent decades, many different research techniques have been used to study rill erosion, as outlined below. However, research into rill erosion in regions with purple soil remains limited.

In 1947, *Ellison (1947)* proposed a conceptual linear feedback model to describe the influence of rill sediment concentration on soil erosion process. *Foster (1982)* distinguished rill erosion from interrill erosion and found that rill erosion led to an increase in eroded sediment particles compared with interrill erosion. Rill erosion is what the water flow scours the ground to form, when the slope thin layer flow assembles the linear small water flow (*Zhao et al., 2015*; *Zhao et al., 2013*). The rills are a product that the slope erosion turns to incising erosion. *Abrahams, Parsons & Hirsh (1992)* subsequently conducted a more detailed and comprehensive study on the mechanisms of rill erosion and the contributing factors. Since then, an increasing number of researchers in China and elsewhere have begun to research rill formation, the characteristics of rill flow hydraulics, and the process of rill erosion.

*Zhang et al. (2002)* designed a series of laboratory experiments to simulate rill erosion and conducted an energetic analysis of the effect of dynamic conditions and rill length on the degree of erosion. *Xiao, Zheng & Jia (2003)* used double flumes (a sand flume and an experimental flume) in runoff plots on an upper slope on the Loess Plateau to study the impact of flow rate and slope gradient on erosion yields in the plateau region. *Yuan et al. (2010)* studied rill runoff and sediment transport on loess slopes with constant-flow artificial drainage combined with rainfall simulation. *Zhao et al. (2014a)* also studied rill runoff and sediment transport on loess slopes but used laboratory experiments that simulated runoff scouring to infer a computational formula for runoff and sediment-transport rate.

*Casalí et al. (2006)* advanced a volumetric method that estimates rill volume, and hence erosion, from a series of cross-sectional areas along the eroded rill, on the assumption that the eroded rill volume is equal to the volume of eroded soil. They highlighted that changes in rill size and morphology can introduce measurement errors with this method. An alternative method for measuring rill volume is to refill the rill with soil, tiny foam particles, rice grains, or other such materials (*Zheng, 1989*). However, in these previous studies, the rills were not well defined; rills of various sizes could not be identified easily, resulting in random and significant measurement errors. In recent years, some researchers have applied the rare-earth-elements (REE) tracing method to investigate the temporal and spatial distribution of rill erosion (*Zhang, Lei & Zhao, 2009*; *Yan et al., 2009*; *Miao et al., 2011*). This method successfully quantifies rill erosion but it is

time-consuming and requires specialized and expensive equipment. In addition, the results of many researchers indicated that various land use types have significant impact on the soil erosion. The change of land use type also caused the changing of soil erosion (*Liu et al., 2014*). Under the different land use types, the erosion mechanism of rill erosion needs to be researched.

In China, the study of rill erosion and sediment transport has concentrated predominantly on the Loess Plateau. However, regions with purple soil have similar hydraulic erosion. The purple-soil regions urgently require investigation, but related research is inadequate and lacks systematic methodology. To date, researchers have adopted the use of artificial rainfall simulations (*Yan et al., 2010*; *Geng, Zheng & Liu, 2010*; *Gao et al., 2014a*; *Gao et al., 2014b*; *Gao et al., 2012*), but quantitative research on the process of rill erosion in purple soil is still lacking.

In this study, we measured the amount of soil erosion along the rill length with the volume-replacement method and calculated the sediment concentration along the length of the rill. Our experimental flume was 12 m long so we could easily observe the sediment transport from rill erosion even with a gentle slope and slow flow. We fit the experimental results to a model describing the relationship between sediment concentration and rill length.

## THE VOLUME-REPLACEMENT METHOD FOR DETERMINING RILL EROSION

Rill erosion is a critical factor underlying the high sediment concentration in water flows on sloping land. In a model situation, sediment concentration increases rapidly at the inlet of a rill and then the rate of increase subsequently slows. Once the sediment concentration reaches a certain threshold, it stops increasing and rill sediment concentration remains stable. At this point, the process in the rill switches from erosion to sediment transport. If sediment concentration continues to increase and exceeds the sediment transport capacity, sediment deposition occurs and can sometimes be observed in rills at a particular distance down the rill. Following sediment deposition, the sediment concentration decreases and new erosion can occur downstream. Therefore, the rate of rill erosion and sediment concentration estimated from the accumulated erosion amounts along the rill length may be higher than the real values. Cycles of erosion and deposition appear randomly, with erosion and deposition alternating both spatially and temporally.

Considering this, the basic assumptions of the volume-replacement method for determining rill erosion are as follows:

(1) The morphology of the eroded rill and the slope of the gully bed remain constant during the measurement; that is, the influence of changing rill morphology on erosion is ignored.

(2) Rill water flow is also assumed to be stable over time, so erosion rates along the rill length do not change with time.

With the above assumptions, the dynamic process underlying rill erosion is as follows: At the rill inlet, the initial part of the rill flow, sediment concentration is zero. As erosion takes place, sediment concentration increases, leading to a reduction in erosion along the rill length. While there is no deposition, sediment concentration shows a net increase. When the sediment particles begin to deposit, erosion alternates randomly with deposition. Sediment concentration at the rill outlet is the maximum potential sediment concentration, as determined by the sediment transport capacity of the flow. It similarly represents the average sediment concentration in time and space of this section of the rill. Thus, the distribution of rill erosion is influenced mainly by the initial section in which sediment content sharply increases (the major of net growth), and sediment content along the rill is influenced mainly by the sediment export concentration.

Given the above, the process of rill erosion under different hydraulic conditions was determined experimentally. Rill erosion was conducted under constant hydraulic conditions to establish the changes in morphology along the rill. The total amount of water used for the rill erosion was calculated from the flow rate and duration. The total soil erosion at different sites along the rill was measured by the volume-replacement method. The average sediment concentration (allowing for deposition as well as erosion) was calculated from the sediment concentration in the collected runoff, and represents the maximum sediment concentration due to rill erosion.

We measured the accumulated amount of erosion in each rill section to obtain the total amount of erosion, and then calculated the corresponding sediment concentration for each section. We could thus obtain an integrated value for sediment concentration along the rill length, despite variations in the rill morphology. Data values that were greater than the export sediment concentration reflect the influence of sediment deposition and were replaced with the outlet sediment concentration.

## MATERIALS AND METHODS

### Experimental materials and soil tank design

Typical purple soil from the Southwest University experimental base for soil and water conservation (106° 25′ 45″E, 29° 49 18″N) was collected for use in this study. The soil contained 38.65% clay content (<0.005 mm), 35.74% silt content (0.005–0.05 mm), and 25.61% sand content. The soil was air-dried before being crushed and passed through an 8 mm square sieve. The experiment was conducted in the rainfall simulation hall at the Institute of Soil and Water Conservation, Chinese Academy of Sciences and Ministry of Water Resources, Yangling, Shanxi Province.

The laboratory flume measured 12 m × 3 m, and the central portion was divided into six 12 m × 0.1 m sections by upright PVC boards (Fig. 1), to imitate well-defined rills and/or to enable water flows to converge and form the required concentrated flow rate. Identical soil materials were glued onto both sides of the PVC boards to imitate the roughness of the soil surface to minimize boundary effects on rill erosion (*Chen et al., 2013*). The bottom 5 cm of the flume was densely packed with clay soil to a bulk density of approximately 1,500 kg m$^{-3}$ to imitate the plow pan layer. Above this layer, 20 cm of flume

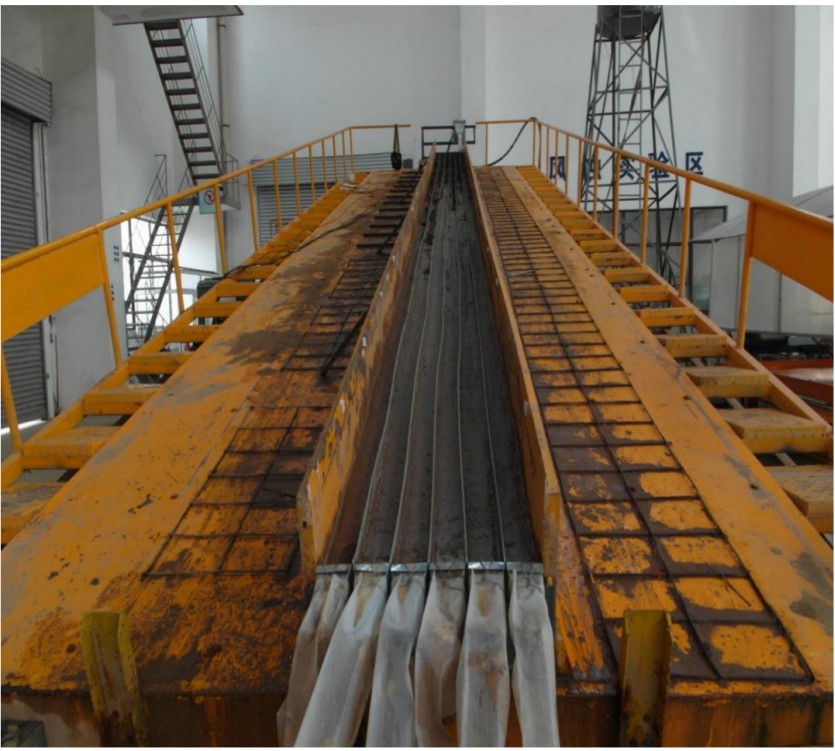

**Figure 1** **Experimental flume.**

was packed with purple soil in layers of about 5 cm to a bulk density of approximately 1,300 kg m$^{-3}$ to simulate the cultivated layer. The soil near the flume walls was packed to be slightly higher than the soil near the middle so that the water flow converged in the middle, which further minimized boundary effects. Prior to the experimental runs, the prepared rills were saturated by running a rainfall simulator for 24 h at an intensity of 60 mm/h to ensure that the water content was close to the field capacity.

## Experimental design

We simulated rill erosion at five different slopes (5°, 10°, 15°, 20°, 25°) and three water flow rates (2 L/min, 4 L/min, 8 L/min, but see also 'Experimental design' below). The choice of flow rate was determined by the critical rainfall intensities that produce rill erosion in sloping croplands on purple soil, converted from rainfall to flow rates by artificial simulation of rainfall intensity (*Yan et al., 2011*; *Olson, Beavers & Fan, 1989*; *Zhang, Lei & Zhao, 2008*; *Lei & Nearing, 2000*; *Gao et al., 2014a*; *Gao et al., 2014b*). Experimental water flow was controlled with an adjustable peristaltic pump. An additional, specially designed, device was used at the rill flow inlet to accelerate the water flow to the required velocity level. Approximately 0.2 m of gauze cloth was placed on the soil surface at the rill inlet to protect the rill surface from being directly scoured by the water flow.

## Experimental methods

All sediment samples were collected in a sampling bucket placed at the rill outlet. After a period of water scouring and rill erosion, the soil flume was adjusted to the horizontal. Plastic film was twined in multiple layers to form eleven thin, waterproof; baffle plates the same width as the rill. The baffles were inserted at 0.5 m, 1 m, 2 m, 3 m, 4 m, 5 m, 6 m, 7 m, 8 m, 10 m, and 12 m from the rill entrance, thus dividing the eroded rill into the following sections: 0–0.5 m, 0.5–1 m, 1–2 m, 2–3 m, 3–4 m, 4–5 m, 5–6 m, 6–7 m, 7–8 m, 8–10 m and 10–12 m. The baffles were deep enough to prevent water from flowing between the rill sections.

Each section of rill was then filled with water and the soil erosion volume was calculated by recording the volume of water in each rill section; then the quantity of soil erosion was calculated from the soil bulk. Unlike particulate matter such as soil, water fills the rill with no gaps. However, the impact of soil pores on the results can be excluded because the soil was pre-saturated. We obtained the values for rill erosion and sediment transport along the entire rill length from the cumulative values for each section.

Rill erosion proved to be low with a slope gradient of 5° and a flow rate of 8 L/min, so the lower flow rates of 2 L/min and 4 L/min were not tested at this gradient. Similarly, we did not test the 2 L/min flow rate with a 10° slope because erosion was already very low at the 4 L/min rate. All other combinations of gradients and flows rates were tested, resulting in twelve different experimental conditions in total. Each condition was repeated three times, resulting in a total of 108 separate trials.

## Calculation of sediment transport along the rill length

Sediment concentration refers to the dry mass of sediment per unit volume of water:

$$S_{ci} = \frac{1,000 \bullet M_{si}}{q \bullet \Delta t} \qquad (1)$$

$$M_{si} = \sum_{i=1}^{11} V_i \rho_b. \qquad (2)$$

In the formula, $S_{ci}$ is the total sediment content at the end of the $i$th rill section (kg/m$^3$), $M_{si}$ is the cumulative mass of eroded sediment at the end of the $i$th rill section (kg), $q$ is the flow rate (L/min), $\Delta t$ is the duration of each trial (min) (*Zhao et al., 2014a*; *Zhao et al., 2014b*); $V_i$ is the total volume of soil erosion in the $i$th rill section, as measured by the volume replacement method (m$^3$); $\rho_b$ is the soil bulk density (kg/m$^3$). From Eqs. (1) and (2), the sediment concentration can be calculated along the length of the rill, revealing the process of soil erosion along the rill length.

## RESULTS AND DISCUSSION

After observing the overall trends in the experimental data, we used the following model to fit the data and reveal how rill sediment concentration changes with rill length:

$$C = A(1 - e^{-Bx}). \qquad (3)$$

**Table 1 Sediment concentration model parameters obtained under different experimental conditions.**

| Slope gradient/(°) | Flow rate /(L·min$^{-1}$) | Regression parameters | | Coefficient of determination |
|---|---|---|---|---|
| | | $A$ | $B$ | $R^2$ |
| 5 | 8 | 206.56 | 0.25 | 0.98 |
| 10 | 4 | 274.39 | 0.28 | 0.93 |
| | 8 | 296.88 | 0.31 | 0.96 |
| 15 | 2 | 387.31 | 0.41 | 0.99 |
| | 4 | 456.93 | 0.43 | 0.99 |
| | 8 | 532.97 | 0.64 | 0.99 |
| 20 | 2 | 395.84 | 0.64 | 0.98 |
| | 4 | 476.80 | 0.70 | 0.99 |
| | 8 | 548.56 | 0.57 | 0.96 |
| 25 | 2 | 451.71 | 0.41 | 0.99 |
| | 4 | 499.22 | 0.62 | 0.96 |
| | 8 | 595.11 | 0.51 | 0.99 |

$C$ represents the sediment concentration (kg/m$^3$); $x$ represents the rill length (m); $A$ represents the maximum possible sediment concentration in the flow, (kg/m$^3$); $B$ represents the decay rate for sediment concentration with speed of rill length increasing (1/m).

Equation (3) represents a situation in which sediment concentration increases with rill length, but the rate of increase decreases exponentially with distance along the rill, eventually tending towards a stable sediment concentration, $A$. When $B$ increases, the curve of the exponential function becomes steeper for a particular value of $A$, meaning that the maximum sediment concentration $A$ is reached after a shorter distance along the rill. The experimental data fits to the model in Eq. (3) are plotted in Fig. 2, and demonstrate how purple-soil sediment concentration varies along the rill length, for the different slopes and flow rates.

The model parameters obtained under the different experimental conditions are listed in Table 1. The coefficient of determination ($R^2$) was greater than 0.9 in each condition, indicating that all fits to the model were very good. It can be seen from Fig. 2 and Table 1 that the tendency of purple-soil sediment concentration to increase with rill length was similar across conditions. Under conditions of known slope gradient and flow rate, the data followed the model closely: sediment concentration increased along the length of the rill but at a decreasing rate until the sediment concentration tended towards a stable value, and the rate of increase tended to zero.

For slopes of the same gradient, the rate at which sediment concentration increased along the rill length accelerated with increasing flow rates. The maximal sediment concentration increased at higher flow rates, and the distance required to reach the maximal concentration decreased. Similarly, for identical flow rates, the rate at which sediment concentration increased along the rill length generally accelerated with steeper

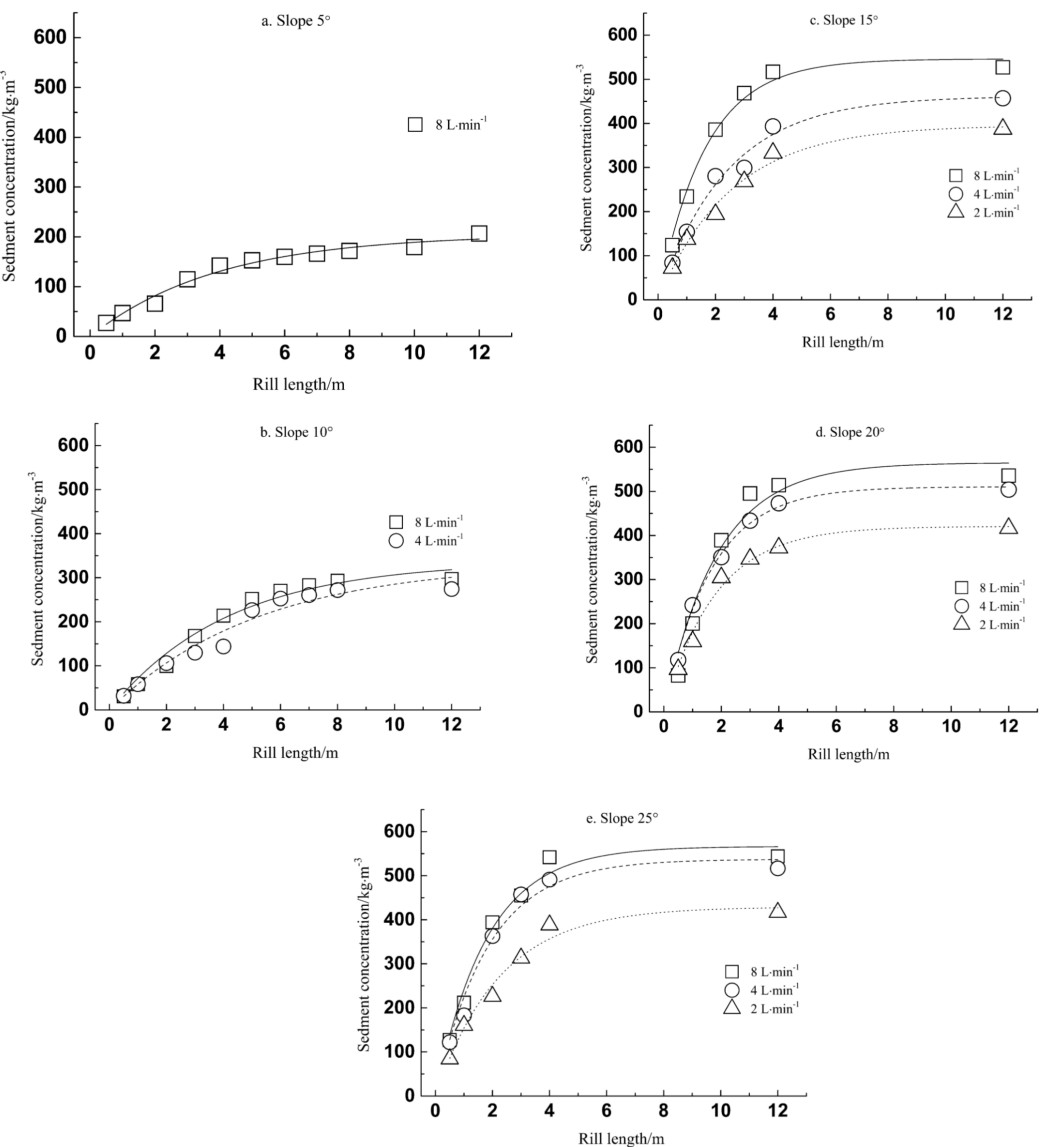

**Figure 2 Sediment concentration as function of rill length.**

slopes. The maximal sediment concentration increased for greater slopes, and the distance required to reach the maximal concentration decreased. Overall, the influence of slope gradient was greater than the influence of flow rate, and the tendency of sediment concentration to increase with rill length was more obvious for changes in slope gradient than flow rate. The significant analysis of slope gradient and flow rate was done. The results showed that the significant analysis values of slope gradient were between 0.01 and 0.05, and the significant analysis values of flow rate were greater than 0.05. Therefore, the influence of slope gradient was more significant than the influence of flow rate. When the underlying surface is the same, the energy in the rill flow is defined by the net flow and current velocity, and the current velocity is determined by the runoff depth and slope gradient. Runoff is the motive power behind soil erosion on slopes. It scours, disperses,

transports, and deposits the soil particles on the soil surface, destroying the soil structure (*Zhang, Lei & Zhao, 2008*; *Cerdà, 1999*; *Brevik et al., 2015*; *Cerdà, 2001*). The frictional force between rill flow and the soil surface influences the susceptibility to runoff scouring. In the context of our experiment, slope gradient affects the soil stress distribution: when the slope was steeper, the water flow dispersed and transported soil particles with greater speed and energy.

Table 1 shows that, for identical slope gradients, parameter *A* increases with increasing flow rates. *A* can be considered to be the maximal potential sediment concentration for purple-soil rill flows. Our results therefore indicate that the maximal sediment concentration for purple-soil rill flow increases with flow rate under conditions of constant slope gradient. Parameter *B* represents the rate at which sediment concentration increases decay along the rill length. It can be observed from Table 1 that *B* tended to increase with both slope gradient and flow rate, indicating a faster rate of increase in purple-soil sediment concentration with steeper slopes and greater flow rates. Similar results have been obtained in previous studies (*Chen et al., 2014*).

## CONCLUSIONS

In this study, we investigated the process of rill erosion along the rill length by using a 12 m soil flume and the volume-replacement method. We used water to backfill the eroded rill because of its mobility and the ease of volume measurement. We ensured that there were no water leakages and so were able to quantitatively measure the process of rill erosion in purple soil. The relationship between sediment concentration and rill length was obtained by fitting the experimental data to a model with two free parameters. The results show that the sediment concentration increases along the length of the rill, and tends towards a stable value. The rate at which sediment concentration increases is highest at the rill entrance and then gradually decreases along the rill length. With steeper slopes and faster flow rates, the increase in sediment concentration is more obvious. These results may provide the basis for understanding the mechanisms of rill erosion and may provide estimates for parameter values in future simulated models of the erosion process in the purple soil.

### Funding

This work was supported by the Foundation of Graduate Research and Innovation in Chongqing under project CYS14054, and Construction Funds for Ecology Key Disciplines for Project 211 Southwest University. The funders had no role in study design, data collection and analysis, decision to publish, or preparation of the manuscript.

### Grant Disclosures

The following grant information was disclosed by the authors:
Foundation of Graduate Research and Innovation in Chongqing: CYS14054.
Construction Funds for Ecology Key Disciplines for Project 211 Southwest University.

## Competing Interests

The authors declare there are no competing interests.

## Author Contributions

- Yuhan Huang conceived and designed the experiments, performed the experiments, analyzed the data, contributed reagents/materials/analysis tools, wrote the paper, prepared figures and/or tables, reviewed drafts of the paper.
- Xiaoyan Chen conceived and designed the experiments, performed the experiments, reviewed drafts of the paper.
- Banglin Luo performed the experiments, contributed reagents/materials/analysis tools.
- Linqiao Ding performed the experiments.
- Chunming Gong contributed reagents/materials/analysis tools, prepared figures and/or tables.

## Supplemental Information

Supplemental information for this article can be found online at http://dx.doi.org/10.7717/peerj.1220#supplemental-information.

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
