# Peer review of "An experimental study of rill sediment delivery in purple soil, using the volume-replacement method"

_PeerJ, doi:10.7717/peerj.1220_

## Round 0.1 · original submission · Major Revisions

· Academic Editor

Major Revisions

I have now received 3 reviews of your manuscript, which are attached for your reference. The reviewers find the paper important and suitable for PeerJ. However, they raise a number of points that need to be addressed before I can consider the paper for publication.

Reviewer #1 suggested referencing some preexisting research in the Section of method and discussion. It is reasonable for you to read the references carefully, and find whether the references are related with your research greatly. Both of the Reviewer #2 and #3 suggested extending the Section Introduction and discussion.

If you wish to revise your manuscript, please take the referee comments fully into account and provide point-by-point responses with a full list of changes.

Reviewer 1 ·

Basic reporting

The submission is fully comply with the all PeerJ polices.

Experimental design

no comments

Validity of the findings

No comments

Additional comments

In this paper, authors measure the amount of soil erosion along the rill length and calculated the sediment concentration along the length of the rill by using simulated rainfall method, which looks interesting. However, I think this manuscript still needs some revision before publication. Some comments as follows:
1. Author should describe how do you measure rainfall intensity by simulated rainfall intensity. There are similar references should be considered, such as Luo et al, 2013, 1-2: 40-47; Sustainability of Water Quality and Ecology; Gao Y, et al. Environmental Monitoring and Assessment, 2010, 171: 539-550; Gao Y, et al. Nutrient Cycling in Agroecosystems, 2009, 263-273
2. There are not any discussion on the erosion mechanism for purple soil, which should focus on the relationship between nutrient loss and soil erosion. Author should also refer some lately research, such as Gao Y et al., Journal of Hydrology, 2014, 517: 447-457; Gao Y et al., Journal of Hydrology, 2014, 511, 692-702; Gao Y et al., Journal of Hydrology, 2012, (420-421): 373-379.
3. In addition, author should supplement a table to calculate the total sediment under different rainfall intensity and compare soil erosion rate. And also need some explanation and discussion.

Reviewer 2 ·

Basic reporting

See below

Experimental design

See below

Validity of the findings

See below

Additional comments

In this manuscript the " volume-replacement method” was used to measure the volume of eroded purple soil under simulated rainfall. According to the authors, the results showed that soil erosion increased with rill length before eventually reaching a maximal value. The increasing rate of soil erosion was greatest at the rill inlet and then gradually slowed down. And slope gradient and flow rate might both result in increases in soil erosion. Through this experimental work, more understanding of rill erosion processes of purple soil (a kind of regosol) were added for the modellers of soil erosion in the region. This study is very interesting and has potential value for the scientific community, although the manuscript will need to be restructured extensively. Therefore my recommendation is that the manuscript is acceptable after major revision.

Specific comments

Line 10, “provide estimates for parameter values for simulated models of the erosion process” could be changed as “provide estimates for parameter values in physical models simulating erosion process”.

Line 78-89, This paragraph could be switch to results section as it comes from the experimental observation. And they also appear in the Abstract as results.

Line 79, Maybe, I am not an expert on soil erosion, I found that although rill erosion is caused by water scouring on the slope, sediment concentration is more meaningful in streams. So, can the authors tell us what happened under flow rate of 8L/min? Does the turbid water flow evidently? Can all the eroded soil particles be transported by the water flow near the upper inlet?

Line 164-165, The sentence is telling that the quantity (not quality) of soil erosion was calculated by the soil bulk multiplied with water volume.

Line 185, Indeed, I feel that the results and discussion section is weak while the methods section is redundant.

Line 186-192, I would suggest that authors should present the results in Fig.2 and fit the data into equation (3). Observation comes first, then modelling.

Line 190-191 and 230, How did you speculate that A can be considered to be the maximal potential sediment concentration for purple-soil rill flows and B represents the decay rate for sediment concentration with speed of rill length increasing (1/m)? Is there any literature to support your speculation or just your inference?

Line 215-217, How did the author find the more important roles of slope gradients than flow rates? By eyes? I would suggest a statistics approach to identify the importance of the parameters.

·

Basic reporting

Dear authors,
the paper is of quality but you need to improve the introduction and discussion with updated citations and references
See attached file

Experimental design

Is of quality

Validity of the findings

The findings are of interest

Additional comments

Dear author
The paper is of quality but you need some improvements with the references

---

## Round 0.2 · accepted · Accept

· Academic Editor

Accept

The authors had made great progresses according to the reviewers' comments. Hence, I am pleased to tell you that your work has now been accepted for publication in PeerJ.